# Evaluating the feasibility and acceptability of virtual group exercise for older adults delivered by trained volunteers: the ImPACt study protocol

Stephen Eu Ruen Lim ,[1,2] Samantha Meredith,[1,2] Samantha Agnew,[3] Esther Clift,[4] Kinda Ibrahim,[1,2] Helen Roberts[1,2]

¹Academic Geriatric Medicine, University of Southampton, Southampton, UK
²NIHR Applied Research Collaboration Wessex, University of Southampton, Southampton, UK
³The Brendoncare Foundation, Winchester, UK
⁴Southern Health NHS Foundation Trust, Southampton, UK

**Correspondence to**
Dr Stephen Eu Ruen Lim;
s.e.lim@soton.ac.uk

## ABSTRACT

**Introduction** Physical activity is important for healthy ageing. Despite strong evidence on the benefits of physical activity for health and well-being, physical inactivity remains a significant problem among older adults. This study aims to determine the feasibility and acceptability of implementing an online volunteer-led group exercise for older adults.

**Methods** A quasi-experimental mixed-methods approach will be used in this study. A training programme will be developed to train volunteers to deliver online group exercises to older adults aged >65 years (n=30). The primary outcome is the feasibility of implementing the intervention. This will be assessed by the number of volunteers recruited, trained, and retained at the end of the study, and the number of exercise sessions delivered and completed by participants. Secondary outcomes include physical activity levels measured using the Community Health Model Activities Programme for Seniors questionnaire, Barthel Index, EQ-5D-5L as a measure of health-related quality of life, SARC-F to determine sarcopenia status, and PRIMSA-7 to determine frailty status. Outcomes will be measured at baseline and at 6 months. Qualitative interviews will be conducted with volunteers(n=5), older adults (n=10) and family members (n=5) to explore their views on the intervention.

**Analysis** Simple descriptive statistics will be used to describe participant characteristics, the feasibility of the study and the impact of the intervention on health outcomes. Parametric(t-test) or non-parametric(Mann-Whitney U test) statistics will be used to analyse continuous variables. $\chi^2$ test will be used for categorical variables. Qualitative data will be analysed using an inductive thematic analysis approach.

**Ethics and dissemination** This study received ethical approval from the University of Southampton Faculty of Medicine Ethics Committee and Research Integrity and Governance committee (ID: 52 967 .A1). Study findings will be made available to service users, voluntary organisations and other researchers who may be interested in implementing the intervention.

**Trial registration number** NCT04672200.

## INTRODUCTION

Physical activity is important for healthy ageing. Some of the benefits of regular physical activity for older people include reduction in falls,[1] better physical functioning[2] and improvement in well-being.[3] The recommended physical activity level for older adults aged 65 years and above is to achieve 150 min of moderate physical activity in a week, with 2 days of strength training.[4] This is often expressed as 30 min of brisk walking or equivalent activity 5 days a week, although 75 min of vigorous intensity activity spread across the week, or a combination of moderate and vigorous activity are also suggested.[5]

Despite strong evidence for the benefits of physical activity, physical inactivity remains a significant problem among older adults worldwide.[6] Studies in the UK have shown that less than 7% of community-dwelling older adults achieve the recommended weekly physical activity level.[7 8] The COVID-19 pandemic has exacerbated the problem of physical inactivity.[9] Rules on social distancing and shielding of vulnerable groups have resulted in increased physical inactivity among older people, with potentially widespread deconditioning.

► This is a novel study exploring the use of trained volunteers to deliver online group exercises for older adults.
► It addresses the issue of physical inactivity among older people, a problem which has been exacerbated by the COVID-19 pandemic.
► A mixed-methods approach will support the development of a toolkit for future implementation.
► For safety reasons, volunteers will deliver seated exercises in online groups.
► The exclusion of older adults who are not able to provide consent limits the generalisability of this study.

The rise of the use of technology during the pandemic may provide a solution to address the problem of physical inactivity among older people. Older adults are embracing modern technology more than ever before and are increasingly familiar with the use of devices and online platforms to stay in touch with their family members and friends.[10] [11] This presents the opportunity to develop online physical activity interventions for older adults to promote increased physical activity. A review of reviews conducted by McGarrigle and Todd found low to moderate evidence that physical activity interventions delivered through the internet and related technologies may be effective in increasing physical activity among older adults in the short term.[12] Online group exercises have been shown to be feasible for older adults.[13] However, there is little evidence on the use of trained volunteers to lead such exercise groups.

Volunteers play an integral part within many health and social care organisations. They enrich patient experience, help deliver high quality services, and support communities in living healthier lives.[14] A report by the King's Fund on volunteering has highlighted that the support volunteers provide can be of particular value to those who rely most heavily on services, such as people with multiple long-term conditions or mental health problems.[15] There is increasing evidence to suggest that with proper training, volunteers can take on more direct roles in supporting older adults particularly with nutrition[16] and mobility.[17]

The Southampton Mobility Volunteer study (n=100, mean age 86 years) showed that volunteers can be trained to deliver chair-based exercises to older adults admitted to hospital.[17] The intervention was shown to be safe, and acceptable to older adults and healthcare professionals, with evidence of improvement in daily step count among patients who received the intervention. The median age of the volunteers in the study was 32 years (interquartile ratio of 17–62 years). A systematic review of 12 studies found some evidence suggesting that volunteer-led physical activity interventions that include resistance exercise training, can improve outcomes of community-dwelling older adults including functional status, frailty status and reduction in fear of falls.[18] However, knowledge gaps still exist on how best to recruit, train, and retain volunteers to deliver physical activity interventions routinely. This study addresses this knowledge gap by exploring facilitators and barriers to the implementation of volunteer-led group exercises for older adults attending social clubs.

Older adults who volunteer to lead physical activity interventions also benefit from their volunteering, with one study showing evidence of improved physical activity levels.[19] Older volunteers also benefit from the social interaction that volunteering brings, with reduced feelings of loneliness and having a sense of achievement.[20] The use of trained volunteers to deliver online physical activity interventions remains an under-researched area and this study addresses the important knowledge gap on the feasibility and acceptability of such interventions.[21]

## Aims

This study aims to determine the feasibility and acceptability of training volunteers to deliver online group exercises to community-dwelling older adults. The specific objectives of this study are to:

1. Develop a training programme for volunteers to lead online group exercises.
2. Determine the feasibility of recruiting, training and retaining volunteers to deliver the online intervention.
3. Determine the acceptability of the online intervention to volunteers, older adults and their carers.
4. Explore barriers and facilitators to the implementation of the online intervention.

## METHODS AND ANALYSIS
### Study design

A quasi-experimental mixed-methods approach will be used to determine the feasibility and acceptability of implementing online volunteer-led exercise sessions in community clubs. The study will be conducted at community-clubs focused on promoting social interaction among older people. Feasibility studies are used to determine whether an intervention is appropriate for further evaluation, to determine sample sizes for controlled trials and to assess whether the ideas and findings can be shaped to be relevant and sustainable. In this study, the feasibility of training volunteers to promote increased physical activity among community-dwelling older adults will be assessed. The acceptability of the intervention will be examined through qualitative interviews to explore the views and experiences of volunteers, older adults participating in the study, and their carers or family members. The impact of the intervention on physical activity levels and functional outcomes will also be measured. This will help inform the sample size calculation for a future effectiveness trial.

### Participant recruitment
#### Volunteer recruitment

The research team will be invited by club staff to attend the virtual clubs to introduce the study. Club volunteers and members who are interested in leading the group exercise will indicate their interest to staff members or approach the research team directly. To lead the group exercise, volunteers must be physically able to perform the exercises themselves and be able to provide written consent. This will be assessed at the training sessions by the trainer (SM).

The participant information sheet and the consent form will be sent through the post. Club members are invited to contact the research team for further discussions about the study. Once the research team has received the signed consent form, volunteers will be invited to participate in the online training. To deliver online group exercise to 30 exercise participants, it is estimated that at least 12 volunteers will be needed to lead 6 online groups. Volunteers will be paired up to lead and deliver the intervention.

## Volunteer training

The volunteer training programme will be led by a member of the research team with a background in sport and exercise science. The training programme was developed based on clinical expertise from therapists and current evidence including experience from a recently completed hospital-based study.[17]

The training will be delivered online through Zoom. Each volunteer will receive two group training sessions, lasting an hour each, and further one to one sessions will be offered to volunteers who wish to have further training. At the first training session, volunteers are given an overview on the importance of physical activity. They are introduced to their role in greater detail and given the opportunity to ask questions. They will be taught the exercise protocol, including the warm up and cool down exercises. The trainer will demonstrate the exercises and provide guidance on how to perform the exercises accurately. They will also be taught the purpose and benefits of each exercise performed. Volunteers will receive training on how to lead the group well, with practical guidance such as being confident, speaking with a clear voice, pacing themselves and providing positive encouragement to the participants. This is then followed by a practical session where the volunteers will practice leading the exercise among the training group members. The training session will also include key aspects such as personal safety and safety of participants. Examples of personal safety include ensuring appropriate clothing and footwear, having good ventilation in the room, staying well-hydrated, and adopting a correct posture while performing the exercises. Volunteers will be trained to coach the participants, ensuring that the exercises are performed safely and effectively to prevent injuries. They will receive training on how to document adverse events that occur during the exercise classes. They will also be given guidance on how to set up their digital devices. Volunteers will receive a training booklet containing the exercises and safety instructions, as well as an exercise video to support their training. Competency assessments based on a checklist will be conducted by the research team to ensure that volunteers are competent in delivering the exercises (online supplemental file 1).

The emphasis of the second training session will be to provide opportunity for the volunteers to practice leading the group sessions. The trainer will observe the volunteers as they lead the exercises and provide feedback to ensure that they are performed accurately. They will also be given advise on how to set up the exercise space to ensure there is adequate space for the exercises to be performed safely, and that the device is optimally set up for participants to see them performing the exercises clearly.

Throughout the study period, fidelity checks will be conducted to ensure that the volunteers are delivering high quality exercise sessions. The fidelity checks will comprise assessment of the volunteer exercise delivery including the safe and effective demonstration and instruction of the warm-up, strength exercise and cool

| Box 1 | Inclusion and exclusion criteria |

**Inclusion criteria**
Older adults aged >65 years.
Able to walk, with or without a walking aid.
Able to provide written consent.

**Exclusion criteria**
Actively participating in other online exercise classes.

down components of the exercises. A member of the research team will attend the group online sessions as an observer to conduct the checks. Volunteers who require further support in delivering the exercises will be offered additional coaching by the research team to ensure that the exercises are delivered correctly.

Regular meetings for the volunteers will be held throughout the study period to provide them with ongoing support from the research team and the organisation running the online clubs. The group meetings will provide an avenue for volunteers to discuss and share their experiences, ask questions and provide suggestions to improve the intervention.

## Exercise participant recruitment

Club members who are interested in participating in the online group exercise will indicate their interest to club staff or approach the research team directly. Participant information sheet and consent forms will be posted to club members.

The inclusion criteria for the group exercise participants are: adults aged 65 years and older, able to walk with or without a walking aid, and able to provide written consent (box 1). Participants who are actively participating in other online exercise classes will be excluded. The aim is to recruit 30 older adults attending social clubs who do not engage in regular exercise.

## Intervention

The intervention consists of a once weekly volunteer-led online group home-based exercise, for a duration of 30 min. The online group will be led by volunteers and hosted on the digital platform Zoom.[22] To comply with the social club's online insurance policy and to minimise the risk of injury to participants, the intervention will only include seated exercises. The seated exercises will focus on strengthening upper and lower limbs and enhancing whole body range of motion and flexibility, tailored specifically to meet the needs of older adults through considering appropriate frequency, intensity, time, type principles and exercise adaptations (eg, in response to joint pain and discomfort). Participants will be encouraged by the volunteers to progress by increasing repetitions and to gently improve range of motion. No supplementary equipment is needed for the online exercises. In addition to the group exercise sessions, participants will also be encouraged to increase mobility and physical activity levels through use of a home activity

booklet, including a physical activity diary. The exercise sessions will be piloted in one club before being extended to other clubs.

## Outcomes measures

Volunteer characteristics such as age, gender, employment status, previous volunteering experience and experience in delivering group exercises will be recorded. Participant characteristics including age, gender, domicile status, body mass index, co-morbidities, cognition (telephone Mini-Mental State Examination) and number of medications will be recorded.

### Primary outcomes

The primary outcome of this study is the feasibility of implementing volunteer-led online group exercise for older adults. This will be assessed by determining:
1. The number of volunteers recruited, trained and retained at the end of the study.
2. The number of physical activity sessions delivered and completed by participants (adherence).
3. The fidelity of the exercise intervention delivered by volunteers.

### Secondary outcomes

The secondary outcome measures include measurement of physical activity levels and performance of activities of daily living (ADLs). Physical activity levels will be measured using the Community Health Model Activities Programme for Seniors (CHAMPS) questionnaire.[23] The questionnaire assesses self-reported weekly frequency and duration of various physical activities typically undertaken by older adults, and is a validated measure of physical activity for older people.[24] Performance of ADL will be measured using the Barthel Index.[25] The Barthel Index measures ADL including transfers, walking, stairs, toilet use, dressing, feeding, bladder control, bowel control, grooming and bathing, to give a maximum total score of 100. It is a validated measure of ADL that is recommended for use in research among older people.[26] TheEuroQol - 5 Dimension (EQ-5D-5L) measure developed by the EuroQol group will be used to determine health-related quality of life.[27] The measure comprises five dimensions: mobility, self-care, usual activities, pain/discomfort, and anxiety/ depression. Each dimension is given a 1-digit score and the digits for the five dimensions are combined unto a 5-digit number that describes the patient's health state. The measure is commonly used in research among older people.[28 29] The Strength, Assistance with walking, Rising from a chair, Climbing stairs, and Falls (SARC-F) questionnaire will be used as a measure to assess for sarcopenia.[30] According to recent guidelines from the European Working Group on Sarcopenia in Older People, SARC-F is the recommended tool for sarcopenia case finding.[31] It comprises five components: strength, assistance in walking, rise from a chair, climbing stairs and falls. The scores range from 0 to 10, with a score of equal to or greater than four being predictive of sarcopenia and

poor outcome. Program of Research to Integrate Services for the Maintenance of Autonomy (PRISMA-7) screening tool will be used to characterise participants' frailty status.[32] The questionnaire contains seven items with a positive score of three or more indicating frailty. The PRISMA-7 tool has been shown to have high accuracy in identifying frail older adults in the community setting.[33]

These outcome measures will be recorded at baseline, using remote assessment formats (eg, telephone based) and repeated at 6 months. Cost analysis will be conducted to determine the cost of implementing this intervention. The analysis will include the cost of training the volunteers, taking into consideration the time needed to deliver the group training sessions and the individual competency assessments, and the provision of training materials.

### Safety and adverse events

To determine the safety of the intervention, adverse events that occur as a result of the intervention will be recorded. Any injuries or symptoms developed directly as a result of the exercises will be recorded as an adverse event. Volunteers will be trained to document any adverse events and report it back to the research team. The clubs also have well established procedures for responding to incidents and accidents including access to emergency contacts of participants.

### Acceptability of the intervention

Interviews will be conducted with older adults, their family members or carers, volunteers, and those involved in recruiting participants and training volunteers, to determine the acceptability of the training and the intervention and to explore barriers and enablers to the implementation of the intervention. Interviews will be semi-structured to help guide conversation while allowing participants the flexibility to elaborate and reflect on any meaningful experiences.[34] The interviews will be conducted by a member of the research team (SM) who is experienced in qualitative interviews. The interviews will be conducted within the first 2 months of the exercise groups to understand how the intervention is being delivered and whether any modifications to the intervention are needed. Further interviews will be conducted towards the end of the study period (at 6 months) to understand stakeholders' lived experience with the intervention and to explore barriers and facilitators for engagement and delivery of the intervention. This will ensure that participants' views are captured during the earlier stages of the intervention, and when the groups are well established.

Older adults will be selected by purposive sampling to share their thoughts and views regarding the implementation of the volunteer-led physical activity session.[35] To ensure that a wide range of views are included in the interviews, participants will be selected to include male and female participants, different clubs, and a representative age range. The aim is to interview at least two participants per club, with a total number of 10 participants. Family

members and carers (n=5) of the exercise participants will also be invited to participate in the interviews. Their views and experience will provide valuable insight into the acceptability of the intervention and the potential barriers and enablers for a future wider implementation. All interviews will be conducted via telephone or online (eg, Zoom, Skype) depending on participant preference.

Semi-structured interviews will also be conducted with volunteers leading the intervention. Discussions among volunteers are likely to generate a wider perspective of the implementation process and may provide insight into the group narrative on their experiences in delivering the intervention and their interaction with the research participants.[36 37] The aim is to interview at least one volunteer from each club, with a total number of five volunteers.

The semi-structured interviews will consist of several key open-ended questions that help define the areas explored but allow the interviewer or interviewee to expand and diverge with the aim of pursuing or developing an idea with more depth. The interview schedules will be underpinned by normalisation process theory (NPT).[38] The interviews will be audiorecorded for data collection purposes.

### Normalisation process theory

The NPT provides a set of sociological tools to understand and explain the social processes through which new or modified practice of thinking, enacting and organising work are operationalised in healthcare and other institutional settings.[38] The use of an implementation theory is useful as it helps researchers identify, describe and explain important elements of the implementation process.[39] NPT was chosen as the framework that underpinned this study as it was systematic in its approach, easy to understand and provided a structured framework to evaluate the implementation process of this study. The online toolkit available for this implementation theory also made the application of NPT convenient and practical.[40] The interview schedules were developed based on the four components of NPT:

1. Coherence: meaning and sense-making.
2. Cognitive participation: this explores the likelihood of all participants being committed and engaged with the intervention.
3. Collective action: the consideration of the work that participants do to make the intervention function.
4. Reflexive monitoring: which involves participants appraising or reflecting on the intervention.

The questions will be framed to explore each component of the NPT in greater detail, with the aim to determine factors that facilitated or inhibited these four key processes which are essential for successful implementation and adoption of the intervention. The first section of the interview guide will focus on interviewees' views on the relevance and importance of physical activity in general, and in particular, the online exercises. Next, participants' views on factors that encouraged or prevented them from being committed and engaged with the online group

exercise will be explored. Volunteers, staff members, older adults, and their carers' experience of the exercise group and how the stakeholders interacted with each other will then be explored to better understand the impact of the group dynamics on the intervention. Interviewees will then be encouraged to reflect on what worked well with the intervention and explore aspects within the intervention that could be improved or require modification.

### Data analysis plan
#### Quantitative data analysis

Data collected will be double entered into a secured database for analysis. Statistical analysis will be conducted using the statistical software SPSS Statistics V.25. Descriptive statistics—median (IQR); mean (SD); number (%)—will be used to analyse the numbers of volunteers recruited, trained, and retained, as well as the type and extent of progression of the seated exercise, and duration of their activity, and patients' adherence to the intervention. Analysis of the above outcome measures will be used to develop an assessment of the feasibility of delivering this intervention. Adherence to the intervention and reasons for non-adherence will be reported.

Outcome measures recorded at baseline will be compared with measurement at 6 months to determine if the PA intervention had an impact on physical activity levels measured by CHAMPS, functional outcomes including performance in ADLs (Barthel Index), quality of life (EQ-5D-5L), frailty status (PRISMA-7) and sarcopenia status (SARC-F). The distribution of each outcome measure will be assessed for normality. Depending on the skew of the data, parametric (t-test) or non-parametric (Mann-Whitney U test) statistics will be used for continuous variables. $\chi^2$ test will be used for categorical variables.

#### Qualitative data analysis

Data collected from the interviews will be transcribed verbatim and analysed with an inductive thematic analysis approach.[41] This approach includes: phase 1—familiarising with the data, phase 2—generating initial codes, phase 3—searching for themes, phase 4—reviewing themes, phase 5—defining and naming themes and phase 6—producing the report. Analysis of qualitative data will be conducted using NVivo. A sample of transcribed text will be read and coded separately and then together by two researchers to create a coding manual that applies to the entire set of transcripts. The codes will be analysed to generate concepts and ideas to determine the acceptability of the intervention, and to identify facilitators and barriers to the implementation process. The codes act as tags or labels to help catalogue key concepts embedded within the raw data. From the codes, themes will be developed to reflect the views and experiences of the community-dwelling older adults and volunteers regarding the online physical activity intervention.

A toolkit that will support a wider implementation of the intervention will be developed. It will be a written manual which contains the training programme,



exercises, competency checklists and information on how best to implement the intervention, taking into consideration the barriers and facilitators determined through the qualitative interviews.

## Patient and public involvement

Patient and public involvement (PPI) involvement was sought in the development of this protocol. Visits were made to two clubs and discussions were held among club volunteers and club attendees about this project. Their views on what type of activities and the frequency of the activities were explored. Club attendees were also asked about what outcomes were important to them, which will be included in the study.

SA who is a coapplicant in this study, will also represent the views of club members and volunteers. She is the head of clubs for the organisation and is experienced in coordinating clubs. She has an overview on how the clubs are conducted and her input on the practical aspects of implementing the intervention will be invaluable. Her practical experience gained through interaction with club members and volunteers will help ensure that their views are represented.

Two additional PPI representatives will be invited to be part of the study management group led by the PPI lead for the Ageing and Dementia theme within the Wessex Applied Research Collaboration (ARC). This group will aim to meet up quarterly to discuss the study progress and will be involved in the design of the research, discussion of the results and dissemination of research findings. PPI has also been involved in the preparation of public-facing materials and study documents.

## Strengths and limitations of the study

This study explores a novel approach of training volunteers to deliver online group exercise to older adults. Application of a mixed-methods approach in this study will help determine the feasibility of the intervention and explore barriers and facilitators to a successful implementation of the intervention. However, a limitation of the study is the lack of a control group. Recruitment of older adults will be through established social clubs, in which recruiting some older adults and excluding others from participating in the exercise groups would be a challenge. Additionally, as this is a feasibility study, we will use a range of social clubs of varying sizes, and client groups with different age ranges and physical abilities, to explore different approaches to the intervention rather than including a control group. Nevertheless, this study has been designed to assess the feasibility and acceptability of the intervention with a view to informing the development of a toolkit to support the wider implementation of the intervention and a future controlled experimental design. This study excluded older adults who are not able to walk. Patients who are bed-bound will not be able to participate in the seated exercises without additional support. This additional support cannot be met by this study. As the intervention is delivered remotely,

participants must have a level of independence to participate safely. Additionally, one of the key outcome measure is physical activity levels and older adults who are bed-bound will have minimal activity levels.

## Ethics and dissemination
### Ethical and safety considerations

This study has received ethical approval from the University of Southampton Faculty of Medicine Ethics Committee and Research Integrity and Governance committee (ID: 52 967.A1). Data will be stored on university repository PURE and will be made available to other researchers on request.

### Dissemination plan

Findings from this study will be published in peer-reviewed journals and presented at national and international scientific platforms. A toolkit will be developed to support future implementation plans.

**Contributors** SERL, EC, SA, KI and HR were involved in the conception and trial design. SM will lead the volunteer training and be involved in data collection, and analysis. All authors contributed in the preparation of this article.

**Funding** The National Institute for Health Research (NIHR) Applied Research Collaboration Wessex funded this research (Award number: not applicable). SERL, SM, KI and HR received support from the NIHR Applied Research Collaboration (ARC) Wessex and the University of Southampton. HR received support from the NIHR Southampton Biomedical Research Centre. The University of Southampton NIHR Academic Clinical Lectureship scheme supported SERL (Award number: CL2018-26-002).

**Disclaimer** The views expressed are those of the authors and not necessarily those of the NHS, the NIHR or the Department of Health.

**Competing interests** None declared.

**Patient consent for publication** Consent obtained directly from patient(s)

**Provenance and peer review** Not commissioned; externally peer reviewed.

**ORCID iD**
Stephen Eu Ruen Lim http://orcid.org/0000-0003-2496-2362

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
