## [Reviewer comments · BMJ Open]

ARTICLE DETAILS

TITLE (PROVISIONAL)	Evaluating the feasibility and acceptability of virtual group exercise for older adults delivered by trained volunteers: the ImPACt study protocol
AUTHORS	Lim, Stephen; Meredith, Samantha; Agnew, Samantha; Clift, Esther; Ibrahim, Kinda; Roberts, Helen

VERSION 1 – REVIEW

REVIEWER	Suzanne Hoi Shan Lo The Chinese University of Hong Kong
REVIEW RETURNED	08-Jul-2021

GENERAL COMMENTS	p.9, line 5, change to 'has received' It is queried if there should be eligibility criteria of the volunteers. It is because the volunteers at least need to be able to physically capable to perform the exercise, or adults. Some screening is needed to select appropriate volunteers to lead the groups. More details are needed for items covered in competency assessments for volunteers and fidelity check mentioned on p.9. Would be helpful to add details about how to train the volunteers e.g. by whom, duration of training, contents of training. Safety precautions for the older adults practising the exercise at home should be mentioned, and how to ensure the safety of the older adults. More details are needed about how the intervention will be conducted online, and any supplementary equipment is needed. The cost analysis is a bit too brief. It would be helpful to provide more details about what item costs will be calculated in this study protocol. How the interview schedule is informed by the Normalisation Process Theory needs to be elaborated. The aspects to explore mentioned in the current protocol is a bit general and appears not very related to the theory. The second paragraph on p.12 mentioned the use of the theory to underpin the study. It would be better to make it clearer about how the theory is used. Is there a control group in the study? Would be helpful to mention how to allocate participants into two groups, and what the control group will receive.
---

REVIEWER	M Hassandra University of Thessaly
-----------------	---------------------------------------

REVIEW RETURNED	12-Jul-2021
-------------

GENERAL COMMENTS	This paper presents the protocol of a study which aims to determine the feasibility and acceptability of implementing an online volunteer-led group exercise for older adults. In general the manuscript is concise and to the point, with a well organized content. My suggestions for further improvement are:  1. I believe there is a limitation that comes out of the study design choice which is good to be on the limitations list. There you can also explain why a comparator group may be difficult to be found. 2. At the introduction section the explanation of the selected secondary outcomes that you expect to have some changes is insufficient. A short reasoning with supported previous literature explaining every measure / outcome choice you have made to include would significantly improve this protocol. 3. The measurement tools are not sufficiently described 4. The section of measurements' description does not include as a measure the patients' adherence to the training program and this info comes as a surprise to the reader later at the data analysis section. 5. Please describe how the fidelity checks will be conducted to ensure that the volunteers are delivering high quality exercise. 6. Please provide some examples of the Competency assessments list of the volunteers 7. Please describe the form of the toolkit you expect to produce for future use (e.g. a written manual, videos or what else) if you already have an idea about that. 8. I was thinking, while reading, that this supervised training of 30min, once a week for 6 months could also aim to educate older adults to exercise alone (repeat the sessions alone) both during the intervention period and/or after the end of the 6months. Supporting autonomy to exercise for life-long is desirable for all ages and the beneficial outcomes of exercise will remain after the intervention ends (at least for those that will develop the motivation to keep exercising). This way the impact of this great effort would increase. 9. Some minor grammatical and typing errors require a good language check from an English language expert.
--

REVIEWER	M Kritz
REVIEW RETURNED	20-Jul-2021

GENERAL COMMENTS	Thank you for the opportunity to review this protocol entitled "Evaluating the feasibility and acceptability of virtual group exercise for older adults delivered by trained volunteer: the ImPACt study protocol". The authors plan to employ a quasi-experimental mixed-methods approach to determine the feasibility of implementing an online volunteer-led group exercise intervention. Mentioned primary outcomes include recruited volunteers, trained volunteers and retained volunteers, the number of physical activity sessions delivered and completed, and the fidelity of the exercise intervention delivered by the volunteers. Mentioned secondary outcome measures include physical activity levels, physical function, cognitive function, and quality of life, measured at baseline and 6 months. This is an important and under-researched topic. The research has the potential to build on previous research. There is research to suggest that older adults may benefit from volunteer-led physical activity interventions and online exercise groups. The present findings could add understanding on the feasibility and acceptability
---

of volunteer-led online exercise groups in older adults. Results could also inform future trials and intervention design

I enjoyed reading this interesting protocol. Some comments are provided below:

Abstract:

1. Please provide more information on quantitative measures (e.g., physical activity- is it self-reported or device-based? how will physical function be assessed?). Please also consider mentioning other measures that you report in the paper (e.g., cognitive outcomes).
2. I think more detail on the analysis would be helpful- what test will be used to determine the impact of the intervention on the outcomes?

Introduction:

Overall, a clearly formulated introduction. I feel that the introduction would benefit from more evidence and a more developed rationale to highlight the novelty and significance of this study.

Some suggestions are provided below.

3. Page 7- Line 7- please support this statement with references.
4. Page 7- Line 19- please consider using an alternative term to "proven" (see APA) – instead use phrase such as "The evidence suggests" - Please also consider replacing "benefit" with "benefits".
5. Page 7- Line 19-20- "Despite....adults" - Please consider supporting this statement with (more) evidence and indicate if you are referring only to the UK or worldwide- any evidence outside the UK?
6. Page 7- Lines 23-28- I am wondering if there are any scientific papers to support that the pandemic decreased physical activity levels? Supporting literature could strengthen this part.
7. Page 7- Line 48-51- Please add evidence.
8. Page 7- Lines 54-58- Consider strengthening your rationale and reviewing some of the existing literature on the effectiveness of older volunteer-led exercise/physical activity groups. Is there any evidence that older volunteers are effective in delivering exercise/physical activity groups? Is there any evidence that they can be trained to be effective in their role? What does this study add?

Methods and Analysis

Overall, a detailed methods section and the mixed-methods design could help you gain comprehensive understanding on the topic. More information on the statistical analysis (e.g., what tests do the authors intend to use and for what variables?), participants and design would further strengthen this methodology/analysis section. For more specific comments, please see below:

9. Page 9- Line 11- You state that there are no exclusion criteria for volunteers. However, you state that exercise participants need to be able to walk. What is the rationale for not screening volunteers for, e.g., mobility?
10. Page 9- Line 49- I am wondering what is your target group ? Is it physically inactive older adults or - like you state- participants who are not taking part in other online exercise groups? Would that mean you are including participants who are active by participating in other forms of exercise? Or will you focus on sedentary participants? Please clarify.
11. Page 10- Intervention- You state that the exercise classes are

	mainly seated exercises. I am wondering what the rationale is for only including participants who can walk. Presumably, it will not be necessary to walk when performing the exercises? What would be examples of exercises where participants are not sitting? You state that participants will be encouraged to increase mobility levels. 12. Page 10- Lines 21-25- Will the authors assess the physical activity levels of participants at baseline? I suggest clarifying. Grammar and Style Overall, a clear manuscript, but please check grammar, spelling and avoid repetitions of identical sentences - some examples are specified below. 13. Title: please consider rephrasing the title by changing- "delivered by trained volunteer" to "delivered by trained volunteers," 14. Page 7- Line 38- please check punctuation- replace "et al" with "et al." 15. Page 8- Line 48- Please check grammar- ""This will help inform...for the future effectiveness trial"".- there seems to be a word missing. 16. Page 9- Line 22- Please check grammar- ""...will be offered to volunteers who wish (not wishes) to have..."" 17. Page 9-Line 44- Please consider changing ""interests"" to ""interest"" or do you mean additional interests (i.e., other than taking part?) 18. Page 11- Line 26- Please consider changing ""among older adults"" to ""with older adults"". Please also correct typos (e.g., ""their family members or caresr"") 19. Page 11-Line 33- Please state who will conduct the interviews and when they will be conducted (e.g., at the end of the intervention? during the intervention?). 20. Page 11- Lines 47-49 - Please avoid repetitions. - "Interviews....preference"- this exact same sentence is stated on page 11 - Line 58- page 12- Line 5- 21. Page 12- Lines 12-16- this information is also a repetition of page 11- lines 44-47. Please check the text for repetition and consider making this part more concise. 22. Page 13-Lines 9-19- Please provide more detail on examined outcome measures and ensure that the provided information is consistent with information provided earlier. Earlier (page 10), you also mention other outcomes (e.g., a cognitive measure- TMMSE). Will this also be assessed? What parametric and non-parametric statistics will be used- more information would be appreciated. 23. Page 13-Please check grammar- For example- Line 11- "to determine if PA intervention had an impact on..." Overall, this is an interesting protocol for research in an important area. I hope the above feedback is useful for the authors for making a few improvements.
--	--

VERSION 1 – AUTHOR RESPONSE

Reviewer 1	
p.9, line 5, change to 'has received'	Thank you. We have amended this. (Pg 7 paragraph 3)
It is queried if there should be eligibility criteria of the volunteers. It is because the volunteers at least need to be able to physically capable to perform the	Thank you for the comment. We have now included a statement to highlight this. (Pg 7

exercise, or adults. Some screening is needed to select appropriate volunteers to lead the groups.	paragraph 2)
More details are needed for items covered in competency assessments for volunteers and fidelity check mentioned on p.9. Would be helpful to add details about how to train the volunteers e.g. by whom, duration of training, contents of training.	Thank you for the comments. We will include the competency checklist as a supplementary file. The remaining details are highlighted in page 8 under the heading 'Volunteer recruitment' and 'volunteer training' headings. (Pg 8 paragraph 1)
Safety precautions for the older adults practising the exercise at home should be mentioned, and how to ensure the safety of the older adults	Thank you for the comment. We have included further details under the 'volunteer training' section to address this point. (pg 8 paragraph 1)
More details are needed about how the intervention will be conducted online, and any supplementary equipment is needed	Thank you for the comment. We have provided further information under the 'Intervention' section. (Pg 9 paragraph 1)
The cost analysis is a bit too brief. It would be helpful to provide more details about what item costs will be calculated in this study protocol.	Thank you for the comment. We have expanded on this under the 'Secondary outcomes' section. (Pg 10 paragraph 2)
How the interview schedule is informed by the Normalisation Process Theory needs to be elaborated. The aspects to explore mentioned in the current protocol is a bit general and appears not very related to the theory. The second paragraph on p.12 mentioned the use of the theory to underpin the study. It would be better to make it clearer about how the theory is used.	Thank you for your comments. We have now included further details under a new section 'Normalisation Process Theory'. (Pg 12 paragraph 1)
Is there a control group in the study? Would be helpful to mention how to allocate participants into two groups, and what the control group will receive.	Thank you for the comment. There is no control group in this feasibility study.
Reviewer 2	
This paper presents the protocol of a study which aims to determine the feasibility and acceptability of implementing an online volunteer-led group exercise for older adults. In general the manuscript is concise and to the point, with a well organized content.	Thank you very much for your comment.
1. I believe there is a limitation that comes out of the study design choice which is good to be on the limitations list. There you can also explain why a comparator group may be difficult to be found.	Thank you for the comment. We have included an additional section on strengths and limitations of the study. However as this is a feasibility study, we will use a range of social clubs with different sizes and clients with varying age range and physical capabilities to explore different approaches to the intervention rather than including a control

	group.
2. At the introduction section the explanation of the selected secondary outcomes that you expect to have some changes is insufficient. A short reasoning with supported previous literature explaining every measure / outcome choice you have made to include would significantly improve this protocol.	Thank you for the comment. We have now included a brief description of each secondary outcome measure used under the heading 'secondary outcomes' in the methods section. (pg 9 and 10)
3. The measurement tools are not sufficiently described	Thank you for the comment. We have now included a brief description of each secondary outcome measure used. (pg 9 and 10)
4. The section of measurements' description does not include as a measure the patients' adherence to the training program and this info comes as a surprise to the reader later at the data analysis section.	Thank you for your comment. This is highlighted under the primary outcomes section as adherence forms part of the feasibility of the intervention. (pg 9 paragraph 3)
5. Please describe how the fidelity checks will be conducted to ensure that the volunteers are delivering high quality exercise.	Thank you for the comment. We have now included this under the section ' Volunteer training'. (pg 8 paragraph 1)
6. Please provide some examples of the Competency assessments list of the volunteers	Thank you for the comment. We will provide this as supplement.
7. Please describe the form of the toolkit you expect to produce for future use (e.g. a written manual, videos or what else) if you already have an idea about that.	Thank you for the comment. We have elaborated on this point this under the section 'Data Analysis plan'. (pg 13 paragraph 4)
8. I was thinking, while reading, that this supervised training of 30min, once a week for 6 months could also aim to educate older adults to exercise alone (repeat the sessions alone) both during the intervention period and/or after the end of the 6months. Supporting autonomy to exercise for life-long is desirable for all ages and the beneficial outcomes of exercise will remain after the intervention ends (at least for those that will develop the motivation to keep exercising). This way the impact of this great effort would increase	Thank you for the comment. While it is beyond the scope of this study, several participants have self-reported that they have been doing the exercises on their own outside the group exercises.
9. Some minor grammatical and typing errors require a good language check from an English language expert.	Thank you very much for the comment.
Reviewer 3	
Abstract 1. Please provide more information on quantitative measures (e.g., physical activity- is it self-reported or device-based? how will physical function be assessed?). Please also consider mentioning other	Thank you for the comment. We have now included a brief description of each secondary outcome measure used. (pg 2 paragraph 2)

measures that you report in the paper (e.g., cognitive outcomes).	
2. I think more detail on the analysis would be helpful- what test will be used to determine the impact of the intervention on the outcomes?	Thank you for the comment. We have included the statistical tests to be used. (pg 2 paragraph 4)
Introduction 3. Page 7- Line 7- please support this statement with references.	Thank you. We have provided 2 references for this statement. (pg 5 paragraph 3)
4. Page 7- Line 19- please consider using an alternative term to ""proven"" (see APA) – instead use phrase such as ""The evidence suggests" " - Please also consider replacing ""benefit"" with ""benefits"".	Thank you for the comment. We have addressed the comment as suggested. (pg 5 paragraph 2)
5. Page 7- Line 19-20- ""Despite....adults" - Please consider supporting this statement with (more) evidence and indicate if you are referring only to the UK or worldwide- any evidence outside the UK?	Thank you for the comment. We have provided further evidence to support this statement. (pg 5 paragraph 2)
6. Page 7- Lines 23-28- I am wondering if there are any scientific papers to support that the pandemic decreased physical activity levels? Supporting literature could strengthen this part.	Thank you for the comment. We have provided a report from Public Health England to support this statement. (pg 5 paragraph 2)
7. Page 7- Line 48-51- Please add evidence.	Thank you for the comment. We have provided a reference to support the statement. (pg 5 paragraph 4)
8. Page 7- Lines 54-58- Consider strengthening your rationale and reviewing some of the existing literature on the effectiveness of older volunteer-led exercise/physical activity groups. Is there any evidence that older volunteers are effective in delivering exercise/physical activity groups? Is there any evidence that they can be trained to be effective in their role? What does this study add?	Thank you for the comment. We have added 2 paragraphs in the introduction section to strengthen the rationale and provide further existing literature on the subject. (pg 6 paragraphs 1 & 2)
Methods and Analysis 9. Page 9- Line 11- You state that there are no exclusion criteria for volunteers. However, you state that exercise participants need to be able to walk. What is the rationale for not screening volunteers for, e.g., mobility?	Thank you for the comment. We have now removed the statement and clarified that volunteers must be physically able to perform and lead the exercise groups. (pg 7 paragraph 2)
10. Page 9- Line 49- I am wondering what is your target group ? Is it physically inactive older adults or - like you state- participants who are not taking part in other online exercise groups? Would that mean you are including participants who are active by participating in other forms of exercise? Or will you	Thank you. The target group are older adults attending social clubs who do not usually engage in regular exercise. We have clarified this in the manuscript.(pg 8 paragraph 4)

focus on sedentary participants? Please clarify.	
11. Page 10- Intervention- You state that the exercise classes are mainly seated exercises. I am wondering what the rationale is for only including participants who can walk. Presumably, it will not be necessary to walk when performing the exercises? What would be examples of exercises where participants are not sitting? You state that participants will be encouraged to increase mobility levels.	Thank you for the comment. Patients who are bed-bound will not be able to participate in the seated exercises without additional support. This additional support cannot be met by this study. As the intervention is delivered remotely, participants must have a level of independence to participate safely. Additionally, one of the key outcome measure is physical activity levels and older adults who are bed-bound will have minimal activity levels.
12. Page 10- Lines 21-25- Will the authors assess the physical activity levels of participants at baseline? I suggest clarifying.	Thank you for the comment. We have now clarified that measurements will be collected at baseline and at 6 months. (pg 10 paragraph 2)
Title: please consider rephrasing the title by changing- "delivered by trained volunteer"" to ""delivered by trained volunteers,"	Thank you for the comment. We have now amended this. (pg 1)
14. Page 7- Line 38- please check punctuation- replace "et al" with "et al."	Thank you for the comment. We have amended this. (pg 5 paragraph 3)
15. Page 8- Line 48- Please check grammar- ""This will help inform...for the future effectiveness trial"".- there seems to be a word missing.	Thank you for the comment. We have amended this. (pg 7 paragraph 1)
16. Page 9- Line 22- Please check grammar- ""...will be offered to volunteers who wish (not wishes) to have...""	Thank you for the comment. We have amended this. (pg 8 paragraph 1)
17. Page 9-Line 44- Please consider changing ""interests"" to ""interest"" or do you mean additional interests (i.e., other than taking part?)	Thank you for the comment. We have amended this.(pg 8 paragraph 3)
18. Page 11- Line 26- Please consider changing ""among older adults"" to ""with older adults"". Please also correct typos (e.g., ""their family members or caresr"")	Thank you for the comments. We have amended this. (pg 10 paragraph 4)
19. Page 11-Line 33- Please state who will conduct the interviews and when they will be conducted (e.g., at the end of the intervention? during the intervention?).	Thank you for the comment. We have now included the requested information under the heading 'Acceptability of the intervention'. (pg 11 paragraph 1)
20. Page 11- Lines 47-49 - Please avoid repetitions. - "Interviews....preference"- this exact same sentence is stated on page 11 - Line 58- page 12- Line 5-	Thank you for the helpful comment. This was repeated as it was in reference to different participant groups. We have now clarified that all interviews will be conducted via telephone or online depending on participant preference to avoid repetition. (pg 11 paragraph 2)

21. Page 12- Lines 12-16- this information is also a repetition of page 11- lines 44-47. Please check the text for repetition and consider making this part more concise	Thank you for the comment. We have removed the statement to avoid repetition.
22. Page 13-Lines 9-19- Please provide more detail on examined outcome measures and ensure that the provided information is consistent with information provided earlier. Earlier (page 10), you also mention other outcomes (e.g., a cognitive measure- TMMSE). Will this also be assessed? What parametric and non-parametric statistics will be used- more information would be appreciated.	Thank you for the comments. We have provided further details for each outcome measure used. (pg 10 paragraph 1) The parametric and non-parametric statistics that will be used are provided under the 'Data Analysis' section. (pg 13 paragraph 2)
23. Page 13-Please check grammar- For example- Line 11- "to determine if PA intervention had an impact on..."	Thank you for the comment. We have amended this. (pg 13 paragraph 2)

VERSION 2 – REVIEW

REVIEWER	Suzanne Hoi Shan Lo The Chinese University of Hong Kong
REVIEW RETURNED	01-Dec-2021

GENERAL COMMENTS	Thank you for the opportunity to review the revised study protocol. The authors have addressed the necessary parts comprehensively. I have no further comments.
---

REVIEWER	M Hassandra University of Thessaly
REVIEW RETURNED	20-Oct-2021

GENERAL COMMENTS	This protocol paper is well written and describes very well all the planned procedures with explanations and reasoning. Except the training of the volunteers. It looks like a black box. All practical details are well explained, who when, where etc. But the content of the training what exactly and how is not described. There are only the topics of trainings and refers to another paper which again is a dead end. How can someone replicate this training of volunteers with so few information? Stating that the trainer of the volunteers is an expert and has done it before is not enough. If possible please add some more information. The protocol paper is the place where all these info should be. Moreover, there is no completed SPIRIT Checklist uploaded. It is an empty form what is uploaded there. Other than these two issues, the work is excellent!
---

VERSION 2 – AUTHOR RESPONSE

Reviewer 2	
This protocol paper is well written and describes very well all the planned procedures with explanations and reasoning. Except the training of the volunteers. It looks like a black box. All	Thank you very much for the comments. We have now included further information on the volunteer training which can be found on page 8

practical details are well explained, who when, where etc. But the content of the training what exactly and how is not described. There are only the topics of trainings and refers to another paper which again is a dead end. How can someone replicate this training of volunteers with so few information? Stating that the trainer of the volunteers is an expert and has done it before is not enough. If possible please add some more information. The protocol paper is the place where all these info should be.	under the section Volunteer training.
Moreover, there is no completed SPIRIT Checklist uploaded. It is an empty form what is uploaded there.	We have now included a completed SPIRIT Checklist.
Other than these two issues, the work is excellent!	Thank you very much for the positive comment.